# Effect of Hydrolyzed Gallotannin on Growth Performance, Immune Function, and Antioxidant Capacity of Yellow-Feather Broilers

**DOI:** 10.3390/ani12212971

**Published:** 2022-10-28

**Authors:** Yuxin Tong, Ying Lin, Bin Di, Guofeng Yang, Jiayi He, Changkang Wang, Pingting Guo

**Affiliations:** College of Animal Science (College of Bee Science), Fujian Agriculture and Forestry University, Fuzhou 250003, China

**Keywords:** hydrolyzed gallotannin, growth performance, broilers, immune function, antioxidant capacity

## Abstract

**Simple Summary:**

The goal of this study is to investigate the effects of HGT on the growth performance, immune function, and antioxidant capacity of yellow-feather broilers. From our findings, the supplementation of HGT in the diet of broilers has no adverse impact on their growth performance, and the diet supplemented with 450 mg/kg HGT is found to improve immune and antioxidant functions of broilers.

**Abstract:**

Tannins were traditionally considered as anti-nutritional factors in poultry production. Recent studies found that the addition of hydrolyzed gallotannin (HGT) could improve animal health; however, the proper dosage of HGT in chickens’ diet is still unknown. Hence, our study aims to recommend its optimal dose by exploring the effects of HGT from Chinese gallnuts on the growth performance, immune function, and antioxidant capacity of yellow-feather broilers. A total of 288 male yellow-feather broilers (34.10 ± 0.08 g) were randomly allocated to four diet treatments, the basal diet with 0 (CON), 150, 300, and 450 mg/kg HGT for 63 days, respectively, with six replications per treatment and 12 birds per replication. The growth performance, slaughter performance, immune organ index, liver antioxidant-related indicators, and serum immune-related factors were evaluated. Results show that HGT supplementation did not influence the growth performance of broilers, but the diets supplemented with 300 and 450 mg/kg HGT increased the semi-eviscerated rate. Furthermore, HGT increased the content of liver T-AOC and the ratio of GSH/GSSG, which can protect against oxidative damage of birds. Additionally, supplementing HGT raised the contents of serum IL-10, IL-4, IL-6, IgA, and IgM. In conclusion, diet supplemented with 450 mg/kg HGT may be the optimal to the health of yellow-feather broilers on the whole.

## 1. Introduction

Over the past decades, antibiotics have been used as feed additives in poultry production to promote growth performance and gut health [1]. However, with the ban on antibiotics as feed additives, novel functional additives are in urgent need to improve animal production performance, enhance immune function, prevent pathogen invasion, and promote the sustainable development of animal husbandry [2,3].

Tannins are secondary metabolites produced by plants and consist of two forms, hydrolysable and condensed tannins [4]. Hydrolyzed gallotannin (HGT), derived from Chinese gallnuts, is a kind of hydrolyzed tannin acid with a glucose molecule as a central core and 10 gallic acid molecules around it [5,6]. In the past, tannins were considered as anti-nutritional factors that exerted negative effects on feed intake, nutrient digestion, and growth performance of monogastric animals [7]. Nevertheless, reactive oxygen species were reported to decrease due to the higher content of antioxidant enzymes after tannin acid treatment in a recent study [8]. Meanwhile, several studies discovered that tannin acid could activate the production of cytokines and stimulate the nuclear factor-κB (NF-κB) signaling pathway to regulate the host’s immune system [9,10]. In addition, a reduction in *Eimeria* spp. oocyst excretion and an anti-inflammatory effect on the small intestine were observed after dietary supplementation of a mix of chestnut and quebracho tannins in the rabbit diet [11]. As for poultry, Minieri et al. [12] reported that dietary chestnut tannin extract supplementation decreased the content of cholesterol and increased the content of oleic acid in egg yolks. Additionally, a study conducted by Al-Hijazeen et al. [13] revealed that 10 mg/kg tannin acid supplementation in the diet of white-feather broilers could better meat color and retard lipid and protein oxidation of breast muscle during storage. However, there are few studies on the effect of tannins on yellow-feather broilers. The optimal dose of tannins in yellow-feather broiler production remains to be explored. Thus, this study was conducted to investigate the effects of dietary supplementation with HGT on growth performance, slaughter performance, immune function, and antioxidant capacity of yellow-feather broilers, in order to provide a theoretical basis for the application of HGT in broiler production.

## 2. Materials and Methods

### 2.1. Experimental Materials

The one-day-old yellow-feather broilers employed in this research were offered by WENS Foodstuff Group Co., Ltd. (Yunfu, China). The HGT product was provided by Fengjiu Biotechnology Co., Ltd. (Zhangzhou, China) and the content of available HGT is 50%, with rice hull as the carrier. Feedstuffs for the broiler diet were offered by Fujian Jinhualong Feed Co., Ltd. (Fuzhou, China). 

### 2.2. Experimental Design, Growth Performance Measurement, and Feeding Management

The experiment was carried out in the Poultry Experimental Station of Fujian Agriculture and Forestry University. All management and experimental procedures followed the animal care protocols approved by the Fujian Agriculture and Forestry University Animal Care and Use Ethics Committee (Fuzhou, China, approval ID: PZCASFAFU22037). 

A single-factor completely random test was conducted in this research. A total of 288 one-day-old male broilers (34.10 ± 0.08 g) were randomly allocated into four groups: (1) corn–soybean-based diet (CON), (2) CON diet supplemented with 150 mg/kg HGT, (3) CON diet supplemented with 300 mg/kg HGT, and (4) CON diet supplemented with 450 mg/kg HGT, with six replications per group and 12 birds per replication. The doses of HGT were selected according to several previous reports and the recommended dose of the manufacturer [14,15,16]. The experimental period was 63 days, including three feeding phases as follows: the early-growth period from Day 1 to Day 21, the middle-growth period from Day 22 to Day 42, and the late-growth period from Day 43 to Day 63. 

The diet composition and nutrient levels of basal diets are shown in Table 1, and the nutrient levels are calculated based on the information available from the feed producer. Broilers were reared in cages (1562.5 cm^2^/bird) with 23 h of light and 1 h of dark in a day, and ad libitum feeding and water intake. The temperature was 35 °C initially and then lowered by 2–3 °C weekly, gradually decreasing to 22 °C until the end of the trial. Additionally, the poultry units were well-ventilated. Immunization with Infectious Avian Influenza Vaccine on Day 5 and vaccination against Newcastle disease on Day 10 and Day 20 were conducted. Meanwhile, the health and mental status of the chickens were observed daily.

Each replication was weighed on Day 1 and Day 63 and feed intake was recorded every day. Average daily gain average (ADG), daily feed intake (ADFI), and feed conversion ratio (FCR) were calculated for 1–63 d.

### 2.3. Sampling Procedure

Two broilers from each replication were sacrificed after 12-h fasting on Day 63 to detect slaughter performance and immune organ index. One broiler from each replication was chosen to detect the antioxidant and immune function. The blood was collected from the basilic vein, left at room temperature for 2 h, then centrifuged at 1450× *g* for 10 min to collect serum samples, which were stored at −20 °C until analysis. The liver samples were collected and stored at −80 °C for antioxidant-related indicator detection. 

### 2.4. Slaughter Performance

The sacrificed birds were manually dissected to determine dressing percentage, semi-eviscerated yield, eviscerated yield, breast muscle yield, and thigh muscle yield. All indexes were calculated as follows: (1) dressing percentage (%) = dressed weight (g)/live weight (g) × 100%; (2) semi-eviscerated yield (%) = semi-eviscerated weight (g)/live weight (g) × 100%; (3) eviscerated yield (%) = eviscerated weight (g)/live weight (g) × 100%; (4) breast muscle yield = breast muscle weight (g)/eviscerated weight (g) × 100%; and (5) thigh muscle yield (%) = thigh muscle weight (g)/eviscerated weight (g) × 100%.

### 2.5. Immune Organ Index

The immune organs of 63-day-old broilers, such as spleen, thymus, and bursa of Fabricius, were also collected and weighted. The calculation formula of the immune organ index was as follows: immune organs indexes (%) = immune organ weight (g)/live weight (g) × 100%.

### 2.6. Liver Antioxidant-Related Indicators

The liver was isolated on an ice tray, fully homogenized with cold PBS (pH = 7.4), with a weight-to-volume ratio of 1:80, and then centrifuged at 1000× *g* for 20 min at 4 °C. Finally, supernatant was collected for measuring the concentration of total protein using a BCA protein quantitative kit purchased from New cell § Molecular Biotech Co., Ltd. (Suzhou, China). The contents of total antioxidant capacity (T-AOC), superoxide dismutase (SOD), glutathione peroxidase (GSH-PX), glutathione (GSH), catalase (CAT), malondialdehyde (MDA), and oxidized glutathione (GSSG) were determined by available assay kits from Enzymatic Biotechnology Co., Ltd. (Shanghai, China), and the ratio of GSH/GSSG was calculated to evaluate glutathione redox status. The absorbance was detected by the iMark™ Microplate Absorbance Reader (Bio Rad, Hercules, CA, USA).

### 2.7. Serum Immune-Related Factors

The levels of immunoglobulin (IgA and IgM) and the concentration of Interleukin-10 (IL-10), Interleukin-4 (IL-4), Interleukin-6 (IL-6), Interleukin-1β (IL-1β), and Interferon-gamma (IFN-γ) in serum were measured using ELISA kits purchased from Enzymatic Biotechnology Co., Ltd. (Shanghai, China), and the absorbance was detected by the iMark™ Microplate Absorbance Reader (Bio Rad, USA). 

### 2.8. Statistical Analysis

The raw data were preliminarily processed by Excel 2019 software. The experimental data were analyzed via one way-ANOVA analysis. Duncan’s multiple comparison was conducted to analyze the differences between groups by using the SPSS 25.0 software. The linear and quadratic comparisons were applied to determine the dose effect of HGT on broilers. Significant difference exists when *p* ≤ 0.05.

## 3. Results

### 3.1. Growth Performance

The effect of HGT on the growth performance of broilers is presented in Table 2. Compared with the control (CON) group, HGT supplementation in the diet had no effects on the ADFI, ADG, and F/G of broilers. Meanwhile, no obvious effect of HGT treatment on overall survival rate (OSR) was observed.

### 3.2. Slaughter Performance

The effect of HGT on slaughter performance is presented in Table 3. Compared with the CON group, the diet supplemented with 300 or 450 mg/kg HGT significantly increased the semi-eviscerated rate of broilers (*p* ≤ 0.05). Additionally, the semi-eviscerated rate linearly increased with an increasing dietary HGT level (*p* = 0.004). However, dressing percentage, total evisceration yield, leg muscle yield, and breast muscle yield were not significantly different between groups.

### 3.3. Immune Organ Index

As shown in Table 4, a dietary addition of HGT did not affect the index of the spleen, thymus, or bursa of Fabricius.

### 3.4. Liver Antioxidant Function

The effect of HGT on the liver antioxidant function is presented in Table 5. Overall, the 450 mg/kg HGT supplementation in the diet of yellow-feather broilers notably increased the content of liver T-AOC compared with the CON group (*p* ≤ 0.05). HGT supplementations of 150 or 300 mg/kg could obviously increase liver GSH/GSSG (*p* ≤ 0.05). However, there were no special effects of HGT treatment on the contents of liver SOD, GSH-PX, GSH, CAT, and MDA.

### 3.5. Serum Immune-Related Factors

The effect of HGT on serum immune-related factors is presented in Table 6. Compared with the CON group, HGT addition increased the content of serum IL-10 (*p* ≤ 0.05), and IL-10 content linearly increased with an increasing dietary HGT level (*p* = 0.001). Additionally, the contents of serum IL-4, IL-6, IgA, and IgM in birds supplemented with 450 mg/kg HGT were higher than the CON group (*p* ≤ 0.05), and linearly increased with an increasing dietary HGT level (*p* ≤ 0.05).

## 4. Discussion

### 4.1. Effects of Dietary HGT Supplementation on Growth Performance of Broilers

Traditionally, tannins were defined as anti-nutritional factors in poultry production and had negative effects on the digestibility of nutrients and the growth performance of broilers [17]. However, Schiavone et al. [15] reported that chestnut tannin supplementation in the broiler (Cobb 508) diet significantly increased ADG and ADFI, especially in broilers aged from 14 to 35 d, and suggested the optimal dosage of HGT was 2000 mg/kg. Meanwhile, dietary supplementation with 500 mg/kg HGT attenuated the negative effects of coccidiosis on Eimeria-challenged broilers, improved host immunity, and decreased the feed conversion ratio [16]. In our study, dietary supplementation with HGT did not affect the growth performance of yellow-feather broilers from 1 to 63 days of age. Consistent with our research, Rezar et al. [14] discovered that chestnut tannin supplementation in the diet of white-feather broilers had no effect on growth performance and nutrient utilization. To some extent, animal breed, dosage, and plant source of the tannins may account for differences obtained in the results. Up to now, most studies related to tannin application in broiler production focus on white-feather broilers [14,15,16,17]. Yellow-feather broilers are more resistant to stress than white-feather broilers; therefore, they are less sensitive to functional additives. In addition, tannins are a group of polyphenols. The differences in active substances because of the diversity in plant sources and extraction processes of tannins can also result in disparate impacts on growth performance. Hence, multiple animal experiments are necessary to evaluate the efficacy of each kind of tannin when applied in broiler production.

### 4.2. Effects of Dietary HGT Supplementation on Slaughter Performance of Broilers

Slaughter performance is an important index to assess the production performance of livestock and poultry. High abdominal fat content of broilers has a negative effect on the processing of meat products and the purchase desire of consumers [18]. Based on our results, the semi-eviscerated rate of yellow-feather broilers linearly increased with an increasing dietary HGT level. One possible reason is the regulation of HGT on fat metabolism. HGT was reported to reduce serum and hepatic lipid concentrations in rats fed high-fat diets [19], leading to a reduced abdominal fat rate and improved semi-eviscerated rate. Additionally, similar results were presented by the study of Annelisse et al. [20], which reported that the dietary quebracho tannins (a condensed tannin) level was negatively correlated with the head–neck mass, and the slaughter performance was improved after tannin supplementation. Another study also demonstrated that the dietary addition of tannin acid from mature leaves and stems of mulberry leaves significantly increased the breast muscle yield of broilers [18]. Hence, dietary tannic acid supplementation may improve the slaughter performance of broilers to a certain extent.

### 4.3. Effects of Dietary HGT Supplementation on Immune Function of Broilers

Thymus, spleen, and bursa of Fabricius are important immune organs for poultry, and their organ indexes can reflect the growth and development status and immune function to some extent [21]. Both the thymus and bursa of Fabricius are important central immune organs. The thymus plays an important role in the development and maturation of T lymphocytes. It gradually grows with age, then develops and matures in adolescence (2~7 weeks of age) [22,23]. The bursa of Fabricius is an important organ for the development and maturation of B lymphocytes [24,25]. The spleen, as a peripheral immune organ, is vital for mature lymphocytes to settle down, present antigens, and induce immune responses [26]. Brenes et al. [27] previously reported that there was no significant effect of grape seed tannins on immune organ development of white-feather broilers. Similarly, in our study, dietary supplementation with HGT did not affect the spleen index, thymus index, and the index of the bursa of Fabricius in 63-day-old yellow-feather broilers. However, the cytokine pattern in serum was altered due to the addition of HGT.

Cytokines play an important role in regulating immune responses. IL-1β and IFN-γ are proinflammatory cytokines produced by a variety of immune cells [28]. IL-10 is an important anti-inflammatory factor secreted by Treg cells and regulated by immune regulatory factor IL-4 [29]. IL-6 is called as B cell differentiation factor, which can promote the development of immune cells and respond to inflammation reactions [30]. A current study showed that dietary supplementation with persimmon-derived tannins could significantly reduce the excessive mRNA expression levels of IL-1β and TNF-α in mouse bone marrow macrophages (BMDMS) infected with *Mycobacterium*, thereby effectively alleviating the inflammatory response of mice [31]. Furthermore, a previous study showed that dietary supplementation with phenolic compounds suppressed the LPS-induced expressions of pro-inflammatory cytokines IL-1β and IFN-γ through NF-κB and MAPK signaling pathways [32]. Similarly, our study demonstrates that the dietary addition of 450 mg/kg HGT could increase the contents of serum anti-inflammatory cytokines IL-10 and IL-4, thereby regulating the immune response. Nevertheless, Ramah et al. [33] demonstrated that dietary supplementation with 500 mg/kg tannin acid had no effect on the mRNA expression levels of IL-1β, IL-4, and IL-10 in spleen. The reason for the above different results on immune function might be due to the diversity of animal species and organs, which have different immune microenvironments [34,35,36].

Immunoglobulins are a category of immunoactive molecules binding specifically to antigens and removing them by sedimentation and phagocytosis. In the domestic chicken, immunoglobulins mainly include IgG, IgA, and IgM [37]. 250 and 500 mg/kg grape seed condensed tannins are found to inhibit the aflatoxin B1-induced reduction of serum immunoglobulin, then alleviate the adverse effects of aflatoxin B1 on the immune system [38,39]. Moreover, hydrolysable tannins in the diet of weaned piglets could increase the IgM content in serum, thus strengthening the immune function of weaned piglets [40]. Our results also show that the addition of 450 mg/kg HGT to the basal diet significantly increased the contents of serum IgA and IgM in broilers. It is reasonable to assume that dietary supplementation with HGT plays a role in the improvement of immune function in broilers, and more studies are needed to elucidate the underlying regulatory mechanisms further.

### 4.4. Effects of Dietary HGT Supplementation on Liver Antioxidant Capacity of Broilers

The antioxidant capacity can reflect the bodily function to scavenge free radicals accumulated in cells and tissues to avoid oxidative damage [41]. The antioxidant system in our body consists of an enzymatic system and a non-enzymatic system [42]. Our study found that dietary HGT supplementation enhanced the body’s antioxidant capacity by increasing the content of liver T-AOC and the ratio of GSH/GSSG. In agreement with our results, the addition of 1% hydrolytic tannic acid could obviously increase the content of unsaturated fatty acids in chest muscle of broilers suffering from heat stress, and the adverse effect of lipid peroxidation caused by heat stress was weakened, indicating that hydrolytic tannic acid may be used as a biological antioxidant for broilers under heat stress [43]. Additionally, Wang et al. [44] reported that 12 mg/kg of grape seed tannins supplemented in the diet significantly increased the concentration of SOD and decreased the concentration of MDA in the context of oxidative stress induced by *E. tenella* infection.

The antioxidant role of phenolic compounds mainly benefits from their reductive properties, which enable them to act as reducing agents, proton donors, and singlet oxygen quenchers [45]. As a kind of natural polyphenol compound, HGT could effectively scavenge superoxide anion radical and hydrogen peroxide and chelate metal cations compared with common antioxidants, such as vitamin E [46,47]. Furthermore, tannic acid could form more stable phenolic oxygen free radicals by reacting with lipid peroxide free radicals and lipid free radicals, thereby blocking free radical chain reactions [48] and further reducing the degree of oxidative damage by inhibiting the production of large amounts of toxic substances such as MDA, 4-Hydroxynonenal, and 2-Enal [49]. Additionally, as a typical xenobiotic for humans and animals, phenolic compounds are metabolized and rapidly removed from the circulation [50]. Accordingly, there is a huge potential to develop HGT as an antioxidant to prevent oxidative damage in yellow-feather broiler production.

## 5. Conclusions

To conclude, dietary supplementation with 450 mg/kg of HGT had a positive impact on the slaughter performance of yellow-feather broilers. No differences were detected on growth performance; however, some beneficial changes in liver antioxidant capacity and immune function were observed. All the results indicate that HGT might be a new functional additive on yellow-feather broiler production.

## Figures and Tables

**Table 1 animals-12-02971-t001:** Composition and nutrient levels of basal diets (air-dry basis).

Items	Phase 1 (d 1 to d 21)	Phase 2 (d 22 to d 42)	Phase 3 (d 43 to d 63)
Ingredients, %			
Corn	58.61	63.94	72.85
Soybean meal, 46% CP	27.80	21.92	13.30
Expanded soybean	9.00	10.00	10.00
Limestone powder	1.24	1.05	1.00
Dicalcium phosphate	1.89	1.69	1.41
DL-Methionine, 98%	0.16	0.10	0.07
Lysine	0.00	0.00	0.07
Premix ^1^	1.00	1.00	1.00
NaCl	0.30	0.30	0.30
Total	100.00	100.00	100.00
Nutrient levels ^2^			
Metabolizable energy, MJ/kg	12.13	12.42	12.76
Crude protein, %	20.83	19.00	15.99
Calcium, %	1.00	0.87	0.77
Total phosphorus, %	0.68	0.63	0.55
Non-phytate phosphorus, %	0.45	0.42	0.37
Lysine, %	1.10	0.98	0.82
Methionine + Cystine, %	0.85	0.75	0.64

^1^ Nutrient levels of premix in Phase 1 (per kg diet): vitamin A 8000 IU, vitamin D_3_ 2000 IU, vitamin E 20 mg, vitamin K_3_ 1 mg, vitamin B_1_ 2.6 mg, vitamin B_2_ 5.4 mg, vitamin B_6_ 5 mg, vitamin B_12_ 0.02 mg, nicotinic acid 40 mg, pantothenic acid 20 mg, biotin 0.2 mg, folic acid 0.8 mg, choline chloride 1000 mg, Cu 8 mg, Fe 80 mg, Mn 80 mg, Zn 60 mg, I 0.35 mg, Se 0.15 mg; Nutrient levels of premix in Phase 2 (per kg diet): vitamin A 8000 IU, vitamin D_3_ 2000 IU, vitamin E 20 mg, vitamin K_3_ 1 mg, vitamin B_1_ 2.6 mg, vitamin B_2_ 5.4 mg, vitamin B_6_ 5 mg, vitamin B_12_ 0.02 mg, nicotinic acid 35 mg, pantothenic acid 20 mg, biotin 0.2 mg, folic acid 0.8 mg, choline chloride 750 mg, Cu 8 mg, Fe 80 mg, Mn 80 mg, Zn 60 mg, I 0.35 mg, Se 0.15 mg; Nutrient levels of premix in Phase 3 (per kg diet): vitamin A 8000 IU, vitamin D_3_ 2000 IU, vitamin E 20 mg, vitamin K_3_ 1 mg, vitamin B_1_ 2.6 mg, vitamin B_2_ 5.4 mg, vitamin B_6_ 5 mg, vitamin B_12_ 0.02 mg, nicotinic acid 30 mg, pantothenic acid 20 mg, biotin 0.2 mg, folic acid 0.8 mg, choline chloride 500 mg, Cu 8 mg, Fe 80 mg, Mn 80 mg, Zn 60 mg, I 0.35 mg, Se 0.15 mg. ^2^ Nutrient levels were calculated values.

**Table 2 animals-12-02971-t002:** The effect of HGT on growth performance of broilers ^1^.

Items	HGT, mg/kg				SEM	*p*-Value		
	0	150	300	450		ANOVA	Linear	Quadratic
Initial weight, g	34.08	34.11	34.07	34.12	0.017	0.704	0.577	0.938
Final weight, g	1900	1894	1902	1859	13.66	0.684	0.371	0.516
ADFI, g	71.24	71.94	72.20	70.74	0.479	0.739	0.845	0.297
ADG, g	31.10	31.54	31.12	30.39	0.216	0.356	0.236	0.190
F/G	2.30	2.31	2.32	2.33	0.009	0.432	0.109	0.868
OSR, %	100	98.61	98.61	100	0.480	0.582	1.000	0.173

^1^ Values are means of six replicates per treatment with 12 birds each. Mean values within a row with no common superscript differ significantly *(p* ≤ 0.05). ADFI, average daily feed intake; ADG, average daily gain; F/G, ratio of feed to gain; OSR, overall survival rate; SEM, standard error of the mean.

**Table 3 animals-12-02971-t003:** The effect of HGT on slaughter performance of broilers ^1^.

Items	HGT, mg/kg				SEM	*p*-Value		
	0	150	300	450		ANOVA	Linear	Quadratic
Dressing percentage	90.25	90.12	92.03	91.56	0.404	0.252	0.106	0.274
Semi-eviscerated yield	83.11 ^b^	84.47 ^ab^	85.72 ^a^	85.43 ^a^	0.327	0.016	0.004	0.142
Total evisceration yield	69.62	69.77	71.26	70.21	0.364	0.399	0.328	0.453
Leg muscle yield	18.70	19.91	19.26	20.23	0.328	0.371	0.185	0.417
Breast muscle yield	14.93	14.29	14.70	14.25	0.312	0.860	0.571	0.844

^1^ Values are means of six replicates per treatment with two birds each. Mean values within a row with no common superscript differ significantly (*p* ≤ 0.05). SEM, standard error of the mean.

**Table 4 animals-12-02971-t004:** The effects of HGT on immune organ indexes of broilers ^1^.

Items	HGT, mg/kg				SEM	*p*-Value		
	0	150	300	450		ANOVA	Linear	Quadratic
Spleen index	0.124	0.148	0.130	0.126	0.006	0.452	0.790	0.239
Thymus index	0.336	0.300	0.347	0.412	0.028	0.536	0.276	0.355
Index of the bursa of Fabricius	0.256	0.206	0.270	0.208	0.013	0.178	0.470	0.801

^1^ Values are means of six replicates per treatment with two birds each. SEM, standard error of the mean.

**Table 5 animals-12-02971-t005:** The effect of HGT on liver antioxidant function of broilers ^1^.

Items	HGT, mg/kg				SEM	*p*-Value		
	0	150	300	450		ANOVA	Linear	Quadratic
T-AOC, U/mg	17.88 ^b^	17.97 ^b^	19.48 ^b^	23.01 ^a^	0.654	0.010	0.002	0.101
SOD, ng/mg	13.68	12.55	13.23	12.35	0.472	0.827	0.591	0.971
GSH-PX, ng/mg	243.7	227.8	252.7	239.2	9.650	0.846	0.835	0.975
GSH, μg/mg	54.31	48.15	51.53	52.82	1.066	0.211	0.907	0.085
CAT, pg/mg	853.4	830.7	821.0	777.5	18.41	0.558	0.178	0.785
MDA, nmol/mg	10.54	9.556	10.16	9.934	0.160	0.171	0.381	0.225
GSH/GSSG	0.960 ^b^	1.066 ^a^	1.061 ^a^	0.986 ^ab^	0.016	0.027	0.261	0.555

^1^ Values are means of six replicates per treatment with one bird each. Different superscripts in a row means significant difference (*p* ≤ 0.05). T-AOC, Total antioxidant capacity; SOD, Superoxide dismutase; GSH-PX, Glutathione peroxidase; GSH, Glutathione; CAT, Catalase; MDA, Malondialdehyde; GSH/GSSG, reduced glutathione/oxidized glutathione disulfide; SEM, standard error of the mean.

**Table 6 animals-12-02971-t006:** The effect of HGT on serum immune-related factors of broilers^1^.

Items	HGT, mg/kg				SEM	*p*-Value		
	0	150	300	450		ANOVA	Linear	Quadratic
IL-1β, pg/mL	526.0	595.8	619.5	622.8	21.49	0.365	0.116	0.446
IL-10, pg/mL	52.91 ^b^	61.08 ^a^	60.48 ^a^	64.72 ^a^	1.343	0.003	0.001	0.234
IL-4, pg/mL	99.65 ^b^	101.6 ^b^	106.0 ^b^	115.4 ^a^	2.225	0.042	0.008	0.339
IL-6, pg/mL	25.72 ^b^	26.80 ^ab^	31.09 ^ab^	35.55 ^a^	1.280	0.012	0.002	0.412
IFN-γ, pg/mL	48.45	56.96	53.94	55.21	2.043	0.482	0.346	0.353
IgA, μg/mL	335.6 ^b^	331.9 ^b^	335.3 ^b^	382.3 ^a^	6.141	0.002	0.001	0.008
IgM, μg/mL	852.6 ^b^	906.6 ^b^	849.1 ^b^	960.3 ^a^	14.97	0.010	0.022	0.204

^1^ Values are means of six replicates per treatment with one bird each. Different superscripts in a row means significant difference (*p* ≤ 0.05). IL-1β, Interleukin-1β; IL-10, Interleukin-10; IL-4, Interleukin-4; IL-6, Interleukin-6; IFN-γ, Interferon-gamma; IgA, Immunoglobulin A; IgM, Immunoglobulin M; SEM, standard error of the mean.

## Data Availability

The raw datasets used and analyzed during the current study are available from the corresponding author on reasonable request.

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
