# Peer review of "Effect of Hydrolyzed Gallotannin on Growth Performance, Immune Function, and Antioxidant Capacity of Yellow-Feather Broilers"

_animals, 2022, doi:10.3390/ani12212971_

Round 1

Reviewer 1 Report

General comments

The manuscriptEffect of hydrolyzed gallotannin on growth performance and general health indicators of yellow-feather broilersdescribes the use of a hidrolized tannin in broilers and its effects on performace and health. In general, the paper is well written, the abstract summarizes the content clearly, methods are sufficiently described to allow reproduction of the experiments, the results are mainly described in a proper way and the discussion is well structured. In my point of view the manuscript could be publish after minor reviews.

Specific comments

Abstract

I think the authors should improve the methodology in this part. There is no information about it here.

Materials and Methods

2.2 -

I think it is too important to say the cm2/bird or cage dimension

Concerning animal care, may be good add the number of protocol

2.3. - In line 93 the authors must say the anatomic name of the “wing vein”

In line 94 the authors must indicate the centrifuge in force “g”.

2.6 – In line 112 the authors must indicate the centrifuge in force “g”.

In line 115 the authors used a large number of abbreviations that are used for the first time in the text. All them must be preceded by the meaning.

2.8 – In line 127-128 – the “and there is no significant difference when P>0.05” is unnecessary.

Results

Table 3 there is no information on the footer note

In line 191 – the number 11 looks like in a different format.

Reviewer 2 Report

Thank you for your submission to Animals. The manuscript falls within the scope of the journal. Please, see the comments below:

1. The manuscripts needs revisions by an native English speaker. 

2. The captions in some of the tables need to be re-written, for example in Table 2. "The same as below", is not coherent in the middle of the text and does not connect well to the rest of the idea. I believe that you are trying to say that the same applies to all of the other tables. Please add a caption independently to each figure for more clarity. 

3. Discussion: 4.1, In this section, there is no discussion of any of the results obtained in the paper, it mainly repeated some of the introduction. Please, include more details relevant to the results obtained in this manuscript. 

4. Line 218: Are, not were, the entire section 4.3 is written in past tense, the immune system and all of the other factors mentioned are still performing those actions. 

5. Line 266: Suffering

6. Conclusions: The results of the study are interesting in that 450 mg/kg of HGT had a positive impact 283 on slaughter performance and were not detrimental for the bird's performance. There were also some interesting immune  responses and antioxidant capabilities. However, there is not enough data to support the claim that HGT can be a candidate to antibiotics in the diet. 

- Does the control diet contain antibiotic? 

- There is no antibiotic listed in any of the diets, therefore, HGT was not evaluated as an antibiotic alternative. HGT might be considered a feed additive however, not an antibiotic alternative.  

Please explain the connection between HGT as an antibiotic alternative, if any, in the introduction and discussion. If not, please consider modifying the introduction and conclusion to consider HGT as a nutritional feed additive. According to the literature reviewed for this manuscript, another author claimed a reduction in oocysts which might be considered an antibiotic alternative effect. However, oocysts counts or antibacterial activity where not evaluated in the current study to make a claim about the antibiotic effect. 

Reviewer 3 Report

The purpose of this study was to evaluate the effects hydrolyzed gallotannin (HGT) on growth performance, slaughter performance, immune organs’ relative weight, liver antioxidant capacity and serum immune function of yellow-feather broilers. The article’s idea is to show that HGT is an alternative feed additive that can be beneficial for broilers. Several critical parameters have been included in this study, especially concerning the growth performance, antioxidant capacity and immune indexes. Overall, the study is quite interesting. Despite these valuable points, I detected some inconsistencies within the manuscript, that made unviable its publication in this journal.

Following are the suggestions which need to be considered.

General comments

1. To begin with, this manuscript needs a comprehensive grammar check.

2. Title

The author wrote “general health indicators” in title, which is obviously inappropriate, because the growth performance, liver antioxidant capacity and serum immune function alone do not fully reflect the “general health indicators”.

3. Materials and Methods

How authors selected the doses used in this experiment should be described.

4. Results

Table 3 and table 4 shall be supplemented with notes. Besides, line 173 and 184, “1” should be deleted.

5. Discussion

The biological explanation is not sufficiently approached or argued. I suggest explaining in greater depth the findings found.

Reviewer 4 Report

The manuscript is concerned on evaluating the effect of hydrolyzed gallotannin on performance and selected health indicators of yellow-feather broilers. The paper itself is very interesting, as it challenges common claim about the anti-nutritional effects of tannins in monogastric animal production and expand knowledge about their mechanisms of action in the body. However, the introduction itself is too short and needs to be completed on information regarding the antioxidant properties of tannins and more information about their immunomodulatory properties which are considered in this paper. It would be good to mention other scenarios (but only in general, not in terms of experiments focused on their optimal amounts) in which tannins have been used so far (if available).

The methodology is described correctly and most of the information, including the design of the experiment. The only comment concerns the equipment used to determine the antioxidant status. Please specify on which devices were performed the measurements antioxidant-related indicators. For KITs, please provide SKU numbers (if available). In the case of the composition of the diets, please indicate whether it was determined on the basis of the information available from the feed producer or as a result of their analyzes on the basis of AOAC procedures?

The presentation and discussion of the results is clear and their interpretation is correct. In addition, the prepared tables are made carefully and contain all necessary information in the footnotes (abbreviations), allowing to clearly assess the obtained results.

For the discussion section I have one question regarding the lines 226-229 (discussion section). Are there any other studies that can confirm this information?

Round 2

Reviewer 1 Report

Once the authors have accepted all the suggestions, I believe that the work can be published in Animals. In the methodology (2.3; line 101), I expected that the authors to report something like ulnar vein, basilic vein or brachial vein. I strong recomend this insertion, however, this does not prevent the publication of the work.

Author Response

Response to the comments of the Reviewer #1:

Materials and Methods

  1. In line 101, I expected that the authors to report something like ulnar vein, basilic vein or brachial vein.

Response: We have modified the methods according to your suggestion.

Reviewer 2 Report

Title correction: "Effect of Hydrolyzed Gallotannin on Growth Performance, Immune Function, and Antioxidant Capacity of Yellow-feather Broilers. 

Line 12: The goal of this study was to investigate.. remove was conducted. 

Line 45: Removed and, replace with "that".

48: studies instead of researches. Same for line 57, 120. 

50: "is except that" the right term? Maybe additionally or it has also been reported? 

56: that "a" diet supplemented with tannin acid... please include the dose. 

59: Remains

158: Control, instead of CON as least the first time in text. 

186: were

212: , and and optimal dosage of HGT of 2000 mg/kg

219: may account for differences obtained in the results.

Line 222: white-feather broilers, therefore, they are less sensitive to func...

224: the diversity in plant sources..

235: removed might

236: similar results were presented by..

248: What is it? when is the adolescence stage in broilers? 

bursa: organ not place.

The spleen

265: a previous study showed that...

283: Remove given the above. 

289: The antioxidant capacity... also the antioxidant system.

297: Supplemented in the diet 

309: xenobiotic 

310: the circulation

Remove "and". No differences were detected on growth performance, however, some beneficial changes on liver antioxidant capacity and immune  function were observed.

Reviewer 4 Report

Thanks to the authors for their reply. The paper was improved and I don't have major comments in this regard.

With best wishes.

Author Response

Response to the comments of the Reviewer #4:

General comments: Thanks to the authors for their reply. The paper was improved and I don't have major comments in this regard.

Response: We are very grateful to your comments and suggestions, which are very helpful to improve the quality of our article.